# The Crosslinking and Porosity Surface Effects of Photoetching Process on Immobilized Polymer-Based Titanium Dioxide for the Decolorization of Anionic Dye

Siti Raihan Hamzah 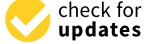, Muhammad Afiq Rosli , Nadiah Sabihah Natar , Nureel Imanina Abdul Ghani, Nur Aien Muhamad, Mohammad Saifulddin Azami , Mohd Azlan Mohd Ishak, Razif Nordin and Wan Izhan Nawawi *

Faculty of Applied Sciences, Universiti Teknologi MARA (UiTM), Cawangan Perlis, Arau 02600, Perlis, Malaysia
* Correspondence: wi_nawawi@uitm.edu.my; Tel.: +604-9882570

**Abstract:** The textile industry is suffering a great challenge regarding wastewater management, primarily due to the implementation of improper systems, specifically for dye wastewater treatment. Photocatalysis is one of approaches that have been used to treat wastewater. Titanium dioxide ($TiO_2$) was immobilized by using the dip-coating technique in this research. Epoxidized natural rubber (ENR) and polyvinyl chloride (PVC) were used as a polymer to bind the $TiO_2$ on the glass substrate. This immobilized $TiO_2$/ENR/PVC underwent a photoetching process at various times to study the crosslink and porosity formations. Reactive red 4 dye was used as a model pollutant for photocatalytic performance. All immobilized $TiO_2$/ENR/PVC samples under 12, 24 and 30 h of photoetching process (TEP12, TEP24 and TEP30 samples, respectively) showed higher photocatalytic activity compared to those without photoetching process (TEP0 sample) due to the intermediate charge in crosslinking reaction after the photoetching process. The TEP24 sample showed the highest photocatalytic degradation; light harvesting; photocatalytic degradation.

**Keywords:** porosity; immobilized $TiO_2$; photoetching; light harvesting; photocatalytic degradation

## 1. Introduction

Photocatalysis is one of the promising techniques among the various AOPs [1]. The most famous semiconductor photocatalyst is titanium dioxide ($TiO_2$) due to its low cost, high stability and non-toxicity compared to other semiconductors [2]. Recently, a normal practice of $TiO_2$ for treating wastewater is in suspension mode as it has [3–5]. However, this practice gives rise to a difficulty in the post-treatment filtration to separate the treated wastewater from $TiO_2$ [6–10]. The post-treatment process can be overcome by developing an immobilized $TiO_2$ that can be recycled and continuously used. There are several methods to immobilize the $TiO_2$ onto a solid substrate, i.e., dip coating from suspension [9–13], electrophoretic deposition [14–16], sol-gel related methods [17–19] and spray coating and sputtering [20–22]. Although the application of immobilized $TiO_2$ has gained much attention in order to solve the problem of the post-treatment process, the low surface area of $TiO_2$ is also considered to cause the decrease in the efficiency of photocatalytic activity [23]. Based on Chen et al. [24], the highest photocatalytic activity is due to the high availability of the active surface area for organic compound absorption. Consequently, the high surface area also can be created by introducing immobilized photocatalysts that can produce a high porosity of the catalyst.

Previously, Tennakone et al. [25] discovered the first technique of immobilization of $TiO_2$ using a polymer as a binder. Their finding has led other researchers to study various types of polymers in immobilizing the $TiO_2$. The addition of a polymer as a binder in immobilized $TiO_2$ is usually applied to improve its adhesion, temperature resisting ability, and durability towards pollutants [26–28], and some polymers are also able to enhance $TiO_2$

photoactivity [29–38]. Generally, solvent-based polymers are used in preparing immobilized $TiO_2$. There are a few polymers that have been used, such as polyethylene sheets [30], thin polyethylene films, polystyrene (PS) beads [31], expanded polystyrene (EPS) beads, ethyl cellulose microspheres [32], fluoropolymer resins, polyethylene terephthalate (PET) bottles [33], polypropylene (PP) granules, cellulose fibers [34], polypropylene fiber (PPF), polyvinyl chloride (PVC) [35], polycarbonate (PC), poly(methyl methacrylate) (PMMA), polyvinyl acetate (PVAc) [36], poly(styrene)-co-poly(4-vinylpyridine) (PSP4VP), rubber latex (an elastic hydrocarbon polymer) [37], parylene and tedlar [38]. These binders have improved the performance of immobilized $TiO_2$ in their own unique characteristics.

Many researchers [7–9,39] have studied immobilizing the $TiO_2$ by using epoxidized natural rubber (ENR-50) and polyvinyl chloride (PVC) as a polymer binder due to their properties of adhesiveness and durability. They state that the ENR compound was photoetched out from the immobilized $TiO_2$ and crosslinking reaction occurred during the photoetching process. However, no study reported the chemical reaction pathway process and the end product formed in the crosslinking reaction during the photoetching process. Hence, this paper focuses on the surface behavior and the photocatalytic activity of immobilized $TiO_2$/ENR/PVC after the photoetching process, suggests a possible reaction mechanism, and presents a crosslinked ENR/PVC end product achieved from the photoetching process.

## 2. Materials and Methods

The titanium dioxide used was commercial Degussa P-25 $TiO_2$ (80 anatase, 20% rutile) nano powder procured from Acros Organics, Geel, Belgium. The polymeric binders applied for the immobilized $TiO_2$ were ENR-50 and PVC supplied by Kumpulan Guthrie Sdn. Bhd., Kuala Lumpur, Malaysia and Petrochemicals (M) Sdn. Bhd., Johor, Malaysia respectively. Toluene, $C_7H_8$ (99.5%) and dichloromethane (DCM), $CH_2Cl_2$ (99.5%) were purchased from R&M Chemicals, Selangor, Malaysia. The model pollutant was reactive red 4 (RR4) dye or known as Cibacron Brilliant Red with 50% dye content (Sigma-Aldrich (M) Sdn. Bhd., Petaling Jaya, Malaysia), as it is widely used in the industry. Throughout this work, ultra-pure water (18.2 M$\Omega$ cm$^{-1}$) was used to prepare all solutions and dilution purposes.

The reflux process of ENR-50 was carried out by using a stirring heating mantle from Fisher Scientific, Gelugor, Penang. The formulation of immobilized $TiO_2$ was homogenized via Crest Ultrasonic cleaner model 4NT-1014-6 (50–60 kHz) from Crest Systems (M) Sdn Bhd, Bayan Lepas, Penang. A 55 W and 65 W compact household fluorescent lamp (Firefly Electric & Lighting Corp., Manila, Philipines) with an emitted UV leakage irradiance of 6.0 W m$^{-2}$ and aquarium pump model NS 7200 (Ace Story Aquatic, Penang, Malaysia) were used as the light and aeration sources, respectively. Two custom-made glass cells with of length 58 mm, width 10 mm and height 80 mm were used for all photocatalytic processes. The support material used for immobilizing the $TiO_2$ was a glass plate (dimensions of 47 mm × 70 mm) with ground surfaces on one side. The concentration of RR4 was determined by using an ultraviolet-visible (UV-vis) spectrophotometer, model DR2000 from HACH Malaysia Sdn. Bhd., Kuala Lumpur, Malaysia. The surface morphology analysis was characterized by a scanning electron microscope (SEM) analyzer model LEO SUPRA 50 VP field emission SEM (Bruker Malaysia Sdn. Bhd., Penang, Malaysia) and 3D optical profilometer (PEMTRON HAWK 3D WT-250, Pemtron, Seoul, South Korea). A Fourier transform infrared spectroscopy (FTIR) analyzer (Perkin-Elmer, model system 2000 FTIR) from Perkin Elmer Sdn. Bhd., Selangor, Malaysia was used to study the changes of functional groups in the treated samples. The binding energy of the chemical compound was determined by performing X-ray photoelectron spectroscopy (XPS) with an Argus detector using an Omicron's DAR 400 dual Mg/Al X-ray source (Thermo Fisher Scientific Malaysia, Selangor, Malaysia). X-ray diffraction (XRD) analysis of the prepared samples was performed by using a Bruker D8 Advance diffractometer (Bruker Malaysia Sdn. Bhd., Penang, Malaysia).

The ENR solution was prepared by refluxing $24.8 \pm 0.05$ g of ENR-50 in 250 mL of toluene at 88–90 °C until all of the ENR-50 was completely dissolved in the toluene and formed an adhesive-like and sticky solution. The preparation of the PVC solution was carried out by dissolving 0.8 g of PVC powder in 35 mL of dichloromethane through sonication for 1 h. Both of these solutions were used to prepare the immobilized $TiO_2$ formulation in which 6.0 g of $TiO_2$ was added slowly into the ENR-50/PVC blend with a mixed ratio of (5:1). Finally, the $TiO_2$ formulation was sonicated by using an ultrasonic machine to completely homogenize the mixture.

A simple dip coating method was applied for immobilizing the $TiO_2$ formulation onto clean glass plates (47 mm × 70 mm, 2 mm thickness). Before the dip coating process, each of the glass plates was dried in the oven at 100 °C for 30 min and weighed by using an analytical balance (Denver Instrument Company, model AA-160) from Alpha Chemicals Sdn. Bhd., Kuala Lumpur, Malaysia. The homogenized $TiO_2$ formulation was poured into a coating glass cell. The glass plate was dipped (cover up to 5.6 cm depth) for 5 s in the formulation and subsequently pulled up manually with a consistent pulling rate. Then, the coated glass plate was dried completely by using an air blower to vaporize the solvent. The weight of the coated glass plate was obtained after scraping off the smooth side of the glass plate. The weight difference between the blank glass plate and coated plate would determine the weight of the immobilized $TiO_2$ and organic binder deposited onto the glass plate. The procedure of coating, drying and weighing was carried out until 2 g of the $TiO_2$ loading was obtained, which typically included three rounds of dip coating. The $TiO_2$ particle loading in the coating was estimated to be around 1.10 to 1.20 g based on the ratio $TiO_2$:ENR:PVC (6:4:1). All of the coated plates were completely dried and stored in plastic dishes before use.

A customized glass cell filled with distilled water (Universiti Teknologi MARA, Perlis, Malaysia) was used to immerse the coated glass plate with aeration air. A 65 W compact fluorescent lamp (Firefly Electric & Lighting Corp., Manila, Philipines) was switched on to initiate the crosslinking reaction. The coated glass plate was photoetched for 1 h. Subsequently, the photoetched coated glass plate was used for the photocatalytic study. The difference of photocatalytic degradation performance between the non-photoetched and photoetched coated glass plates was finally observed.

A 55 W compact fluorescent lamp (Firefly Electric & Lighting Corp., Manila, Philipines) was placed horizontally in front of the custom-made glass cell reactor. A volume of 16 mL of 30 ppm RR4 dye solution was poured into the glass cell. Then, the glass plate was immersed into the glass cell with aeration and a light source. At 15 min time intervals, the degradation of the dye during the photocatalysis process was observed and the absorbance was measured by using a DR2000 spectrophotometer (HACH Malaysia Sdn. Bhd., Kuala Lumpur, Malaysia). The results were converted into ln $C_0/C$, where $C_0$ is the absorbance of the initial concentration and C is the absorbance at any time (*t*), and thus the values were plotted against irradiation or contact time. Based on the Langmuir–Hinshelwood rate model, the slope of the linear line was taken as the pseudo first-order rate constant.

## 3. Results

The surface morphology as seen via field scanning electron microscopy (FESEM) of TEP0, TEP12, TEP24 and TEP30 for immobilized $TiO_2$/ENR/PVC without photoetching and with 12, 24 and 30 h photoetching, respectively, is shown in Figure 1. The morphology of immobilized $TiO_2$/ENR/PVC without photoetching in Figure 1a,b shows a compact surface, which indicates that the $TiO_2$ particles are covered with a layer of ENR/PVC. Figure 1c,d illustrates the immobilized $TiO_2$/ENR/PVC after 12 h of photoetching, which reveals that the fully covered layer is being degraded and turning into a hole with a sponge-like layer structure after a 12 h photoetching process. The ENR/PVC layer diminished from the $TiO_2$ after the 24 h photoetching process (Figure 1e,f) and remained constant until 30 h of the photoetching process (Figure 1g,h). The amounts of ENR/PVC layer and $TiO_2$ that leached out in the water during the photoetching were ca. 0.08 and 0.096 g, respectively.

Based on previous researchers [7–9,40,41], they state that a porous immobilized $TiO_2$ is produced as the ENR-50 leaches out during the 8 to 12 h photoetching process. This finding corresponds with the result obtained and shown in Figure 1a,h, as the layer of ENR/PVC diminished slowly upon time.

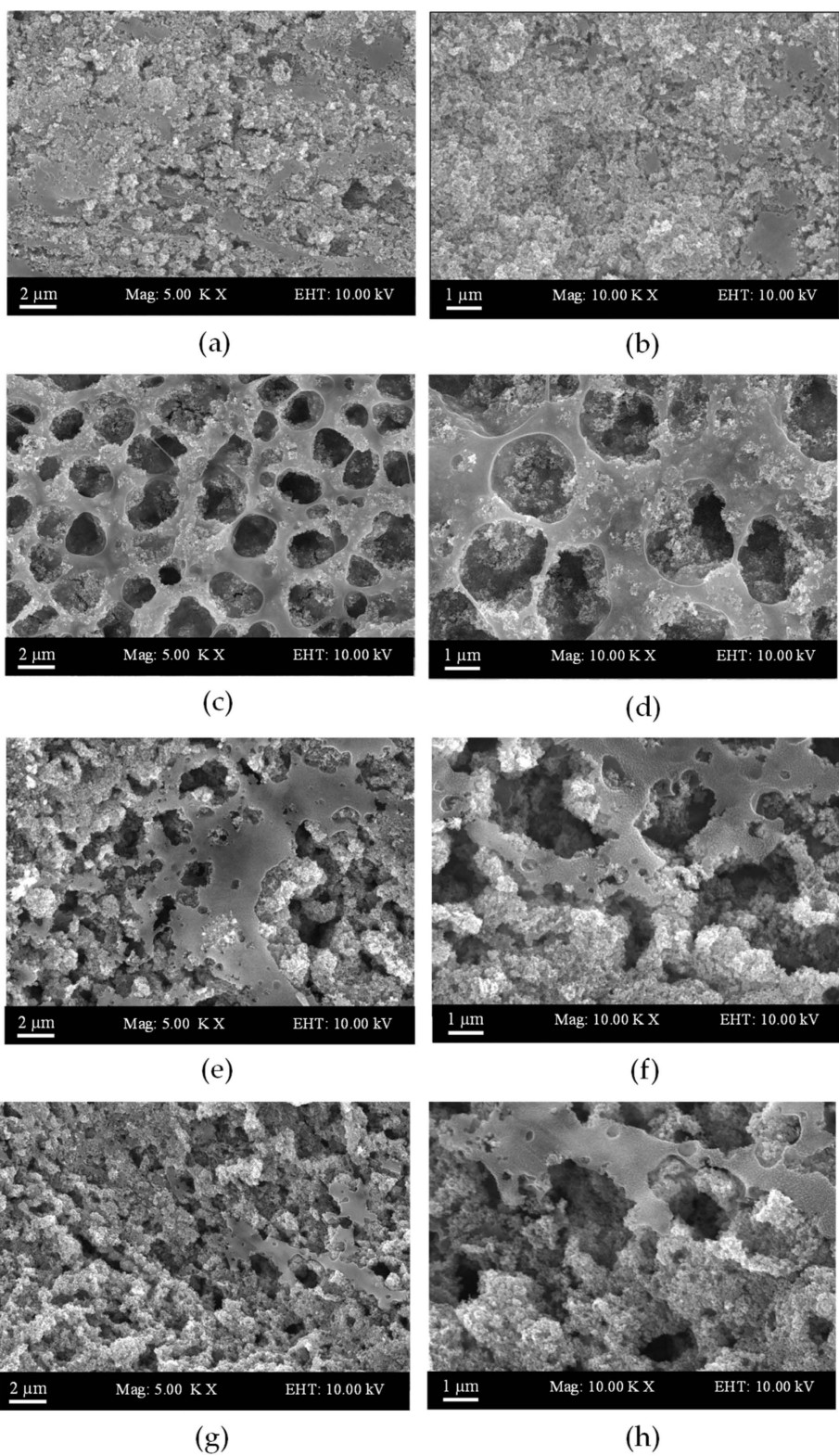

**Figure 1.** Field emission scanning electron microscopy (FESEM) of (**a,b**) TEP0, (**c,d**) TEP12, (**e,f**) TEP24 and (**g,h**) TEP30 under 5000× and 10,000× magnification, respectively.

The porosity transformations in TEP0 and TEP24 are supported by surface topography characterization via 3D optical profilometer (PEMTRON HAWK 3D WT-250, Pemtron, Seoul, South Korea) using an optical probe at 5× magnification, as shown in Figure 2a,d. The height of the surface can be determined by the color indicator shown in the 2D surface profile (Figure 2a,b). Based on the color indicator, the blue color indicates the deepest height while the red color depicts the tallest height. The 3D surface profile illustrates that the surface of TEP24 was more porous compared to TEP0.

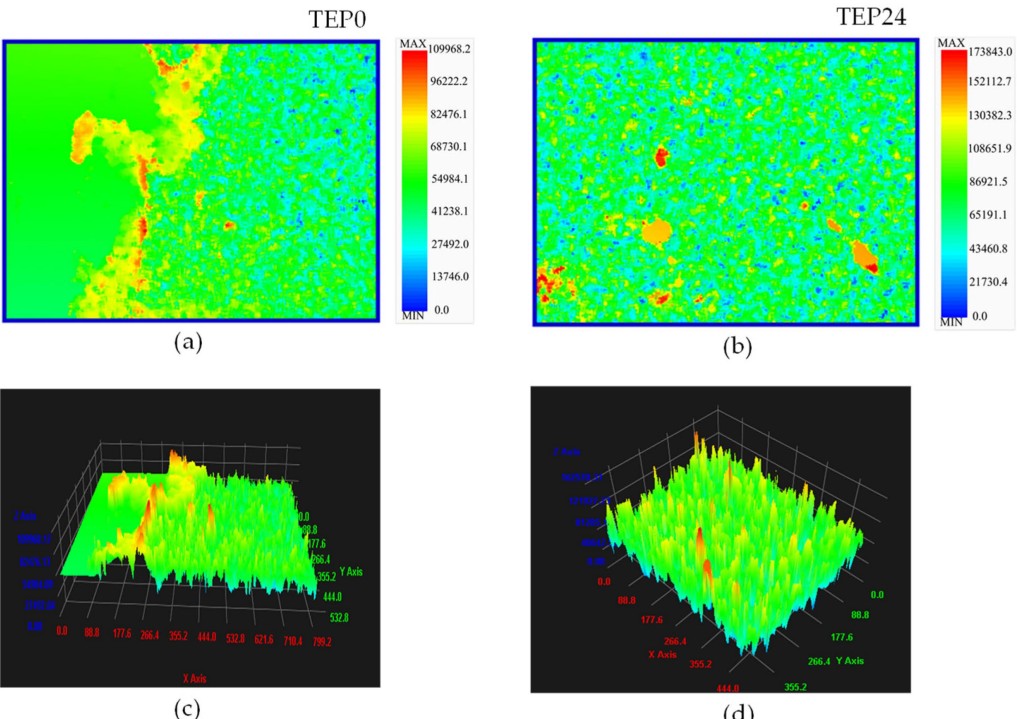

**Figure 2.** (**a**) 2D surface profile and (**b**) 3D surface profile of TEP0 and (**c**) 2D surface profile and (**d**) 3D surface profile of TEP24.

The parameters in the 2D surface roughness category are average roughness ($R_a$) and familiar root mean square roughness ($R_q$). Both of these parameters are given by the following equations [42]:

$$R_a = (1/N) \sum_{J=1}^{N} |Z_J| \tag{1}$$

where $N$ is the total number of measured points and $Z_J$ is the vertical height deviation from the average height of the surface.

$$R_q = \sqrt{\left( \sum (Z_J)^2 \right) / N} \tag{2}$$

where $Z_J$ is the vertical deviation with respect to a mean line for the $j^{\text{th}}$ point of the total $N$ measured points.

The $R_a$ values for TEP0 and TEP24 are 8101.95 and 15,674.81 nm, while the $R_q$ values are 10,449.51 and 19,062.62 nm, respectively. The surface roughness of TEP24 is higher than TEP0 due to the high porosity as ENR-50 leaches out from the immobilized TiO$_2$ as signified by the FESEM results in Figure 1.

The XRD analysis for the TEP0 and TEP24 samples was performed on a Bruker D8 advance diffractometer (DKHS Sdn. Bhd., Petaling Jaya, Malaysia) operating in the reflection mode with Cu-Kα radiation (35 kV, 30 mA) and a diffracted beam monochromator (DKHS Sdn. Bhd., Petaling Jaya, Malaysia), using a step scan mode with the step of 0.075° (2θ) and 4 s per step. The XRD patterns of TEP0 and TEP24 are shown in Figure 3. All

peaks were detected as anatase phase compared with JCPDS Card No: 01-071-1169. The positions of 2θ at 25.31°, 26.95°, 37.79°, 38.57°, 48.04°, 53.89°, 55.07°, 68.76°, and 70.30° correspond to the Miller indices of (101), (103), (004), (112), (200), (105), (211), (116), (220), and (215) plane, respectively. Assuming that the crystalline powder did not align itself to a preferential orientation during fixation, the strong diffraction peaks at 25.31° and 48.04° confirm the samples in the anatase structure. The higher peak intensities are due to the etching of the polymer, exposing more $TiO_2$ crystals to the X-ray probe. The morphology of the ceramic powder is unlikely to change under the reported experimental condition, other than some electrochemical etching.

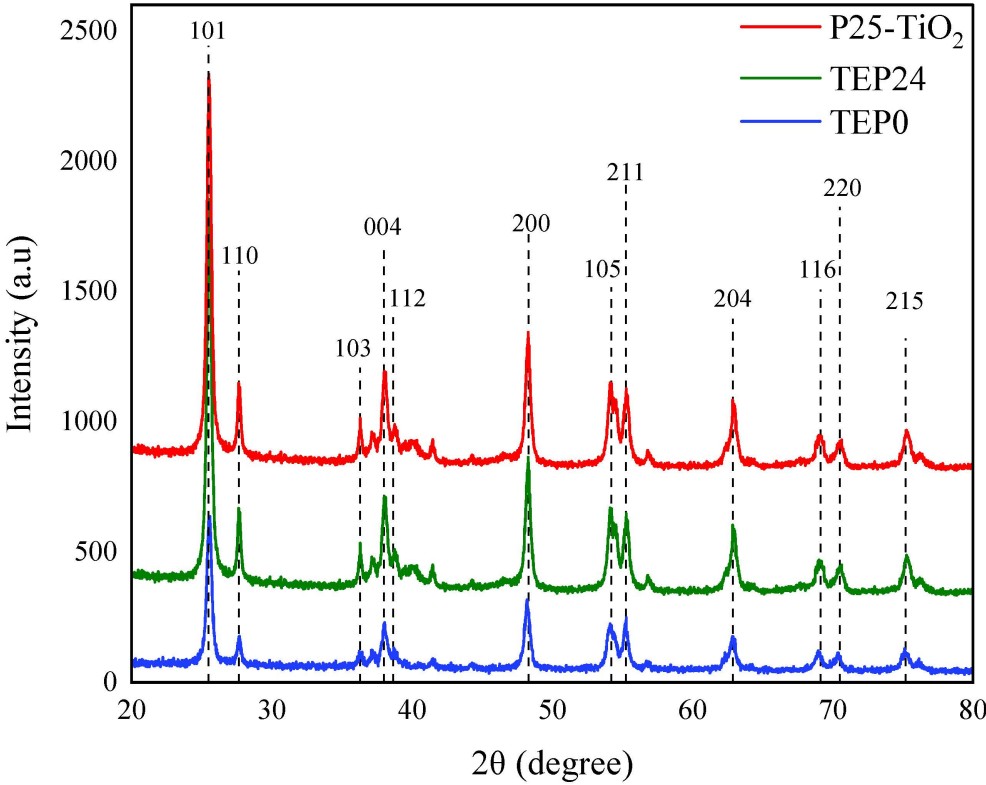

**Figure 3.** X-ray diffraction patterns of TEP0 and TEP24.

Table 1 shows that the crystallinity index (CI) values of TEP0 and TEP24 are 51.21% and 57.30%, respectively. The CI value for TEP24 is 6.07%, which is higher than the TEP0 sample due to the ENR/PVC decrement, as also observed in the FESEM images. Based on the XRD pattern, the crystallinity index (%) can be calculated via XRD data and origin software using the following equation [43–45]:

$$\text{Crystallinity index (CI)} = \text{Area CP/Area CAP} \tag{3}$$

where area CP is the area of all the crystalline peaks while area CAP is the area of all the crystalline and amorphous peaks.

**Table 1.** Crystallinity index (CI) of TEP0 and TEP24.

| Sample | Crystallinity Index (%) |
|---|---|
| TEP0 | 51.21 |
| TEP24 | 57.30 |

Figure 4 depicts an optimized slab model of the $TiO_2$ structure that has been calculated before the calculations of the electronic structures. The slab model was executed to acquire

the lattice parameters with the lowest energy. The computational results for the structural parameters such as lattice parameter, volume, and bond length are shown in Table 2. Based on the data obtained, the structural $TiO_2$ can be reported as anatase form due to its tetragonal shape. The volume of the $TiO_2$ between TEP0 and TEP24 has no significant difference, which indicates that there is no structural alteration during the photoetching process.

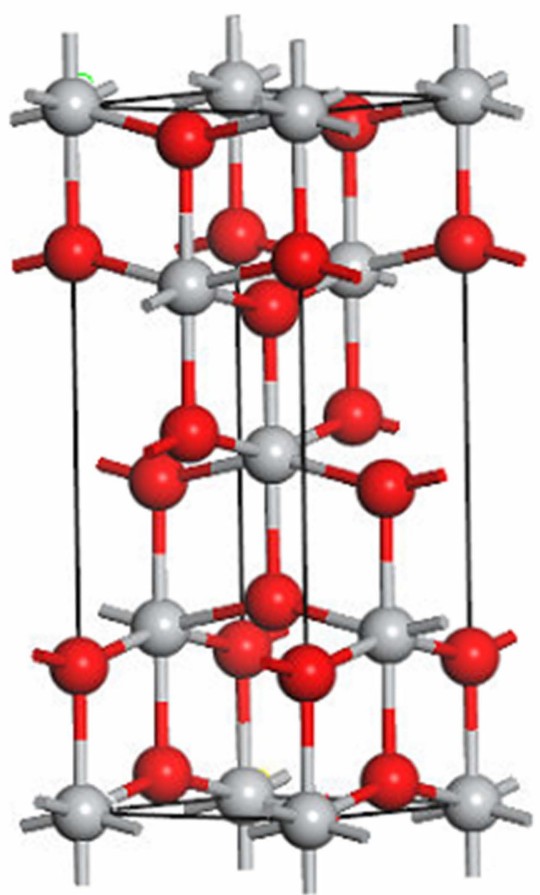

**Figure 4.** Structure of $TiO_2$. The grey and red balls indicate Ti and O atoms, respectively.

**Table 2.** Lattice parameter, volume, and bond length of TEP24 and TEP0.

|  | Lattice Parameter (Å) | Volume (Å³) | Bond Length (Å) | |
|---|---|---|---|---|
| TEP24 | a = 3.7752 | 136.4475 | Ti(1)-O | 1.9310 |
|  | b = 3.7752 |  | Ti(2)-O | 1.9861 |
|  | c = 9.5742 |  | O(1)-O | 2.4608 |
|  |  |  | O(2)-O | 2.7910 |
| TEP0 | a = 3.7758 | 136.3951 | Ti(1)-O | 1.93102 |
|  | b = 3.7758 |  | Ti(2)-O | 1.98602 |
|  | c = 9.5670 |  | O(1)-O | 2.46201 |
|  |  |  | O(2)-O | 2.79050 |

The electronic structures of TEP0 and TEP24 were investigated by setting the same k-points mesh to sample the first Brillouin zone for TEP0 and TEP24. Figure 5a,b represents the calculated band gaps of TEP0 and TEP24, which are 3.206 and 3.210 eV, respectively. The conduction band minimum (CBM) is located at M, while the valence band maximum (VBM) is located near G. The band gap was not affected by the photoetching treatment, as the band gap values of both samples are obscure to the theoretical band gap of $TiO_2$, which is 3.20 eV. Nonetheless, the calculated band gaps are overvalued compared with the

experimental value obtained by UV-Vis DRS analysis as shown in Figure 6b. This may be due to the DFT limitation, as the discontinuity in the exchange correlation is not taken into account within the framework of DFT.

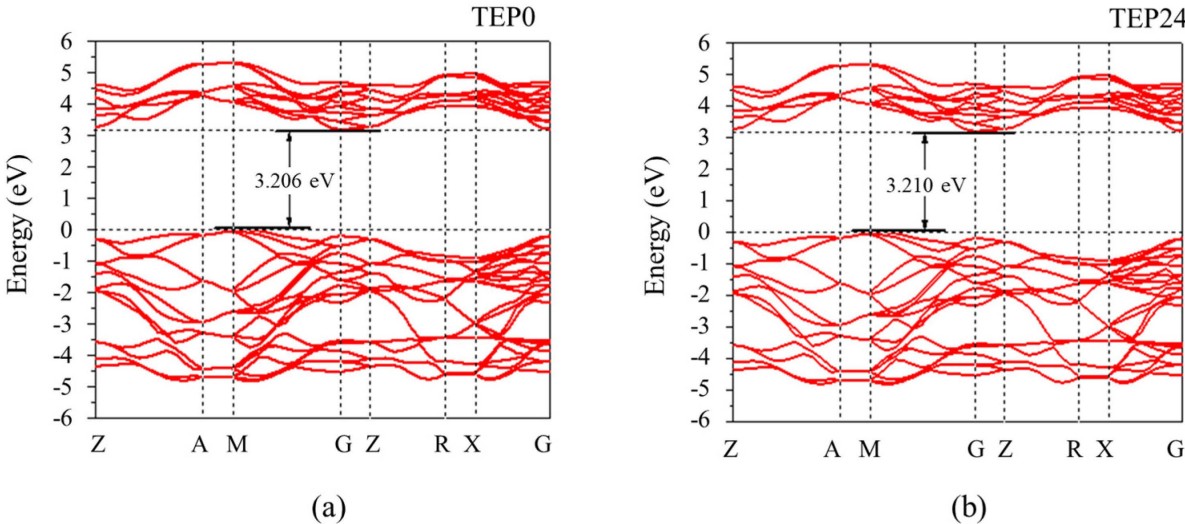

**Figure 5.** Calculated band structure of $TiO_2$ in (**a**) TEP0 and (**b**) TEP24 samples.

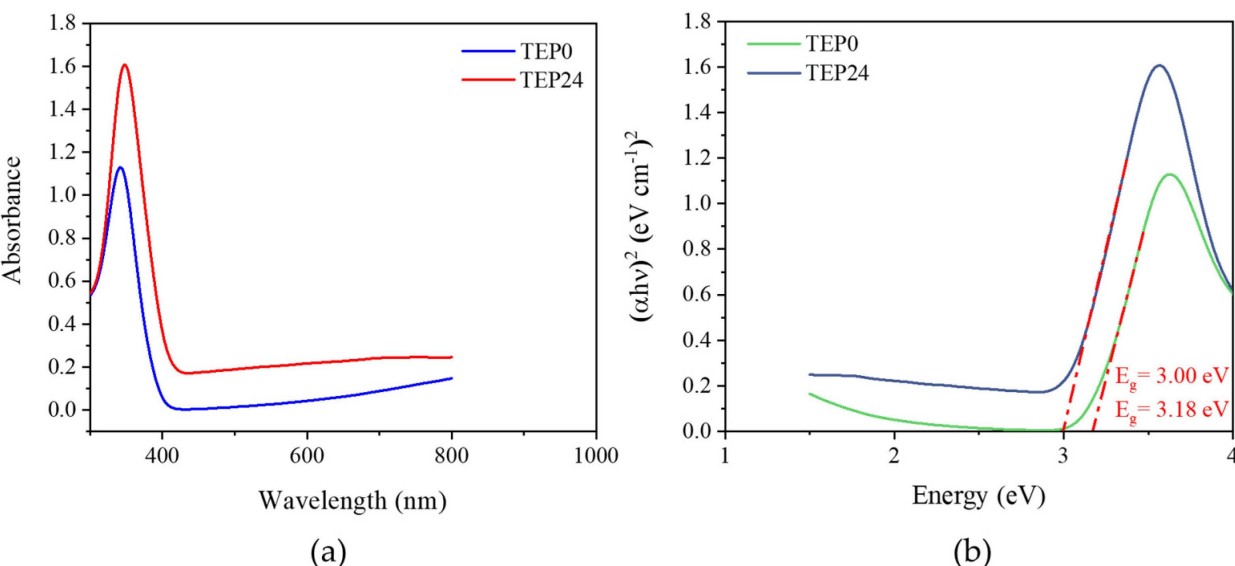

**Figure 6.** (**a**) UV–Vis DRS and (**b**) corresponding Kubelka–Munk function plots of different samples.

Based on Figure 6a, TEP0 exhibits higher absorbance of the maximum peak ($\lambda_{max}$) compared to TEP24. The difference in UV absorption intensity observed is also a consequence of the exposed $TiO_2$ crystals after the binder was photo-etched with the very remote possibility that there might have been some preferential orientation of the crystals during curing. The band gap energy ($E_g$) of the samples was calculated by the Tauc equation $\alpha h\nu = A(h\nu - E_g)^n$, where A is constant, and $\alpha$, h, $\nu$, and $E_g$ are the absorption coefficient, Planck's constant, photon energy, and band gap, respectively [46]. $n$ is defined by the optical transition of a semiconductor ($n = 1$ for $TiO_2$). As shown in Figure 6b, the $E_g$ for TEP0 and TEP24 is approximately 3.18 and 3.00 eV, respectively. The $E_g$ for TEP24 is lower than the TEP0 supporting the reduction of electron hole recombination incidence. Thus, the smaller electron hole recombination after the photoetching process results in a higher photocatalytic degradation.

The total densities of states (TDOS) and partial density of states (PDOS) of TEP0 and TEP24 treated by Gaussian broadening are verified in Figure 7a,b. The band gap can be defined by the separation between the CBM and VBM, which are extending in energy ranges from 0 to 6 eV and from −6 to 0 eV, respectively. The DOS for both samples shows that the VBM is predominantly made up of O 2p states while the CBM is mostly made up of Ti 3d states alongside a minimal amount of O 2p states. These results are consistent with other DFT calculations on TiO$_2$ polymorphs [47].

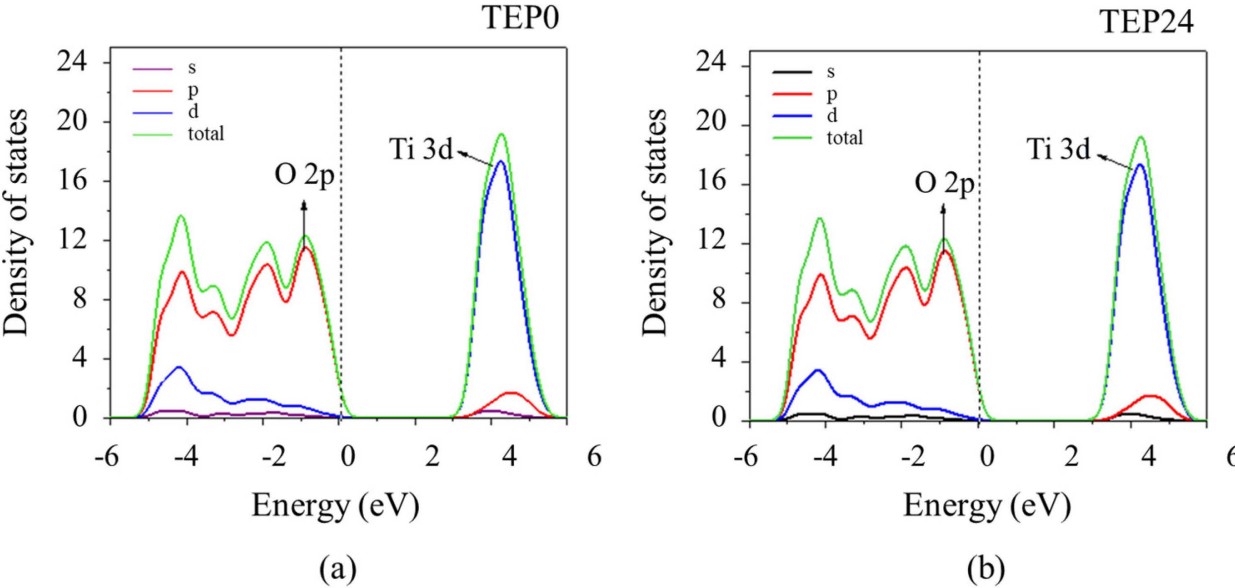

**Figure 7.** Density of states for (**a**) TEP0 and (**b**) TEP24. Green solid lines: TDOS, red solid lines: O 2p states, and blue solid lines: Ti 3d states. The black dashed line represents the position of the Fermi level.

The FTIR spectra of TEP0, TEP24, and Degussa P-25 TiO$_2$ are shown in Figure 8. The peaks at 1251, 1332, and 1432 cm$^{-1}$ in Figure 8 indicate the presence of PVC, while the peak at 1432 cm$^{-1}$ indicates the presence of the ENR-50 compound [7–9,41]. The destruction of the CH$_2$Cl deformation in PVC compounds and the reduction of ENR-50 compounds can be seen by the decrease of peak intensity at 1251 and 1432 cm$^{-1}$, respectively. This is due to the opening ring of ENR-50 and crosslinking reaction between ENR-50 and PVC during the photoetching process as pointed out by [7–9]. Moreover, there are two new peaks of the C–O bond in the TEP24 spectrum at 1167 and 1062 cm$^{-1}$, which confirms the formation of aliphatic ether due to the crosslinking reaction. Hence, even though the porosity formed due to the leaching out of ENR, there is some of the remaining ENR turning into a crosslinking reaction as shown in the possible reaction mechanism as depicted in Figure 10. Instead of a crosslinking reaction occurring, there is some potential that the PVC may be converted into a polyenes precursor, which can serve as a photo-sensitizer, thus enhancing the photocatalytic activity of TiO$_2$ [8].

X-ray photoelectron spectroscopy (XPS) was performed to confirm the binding energy of the elements detected in TEP24. The crosslinking reaction was successfully produced by the photoetching process. Figure 9a,b demonstrates the wide, C1s, O1s, and Cl2p spectra for TEP24. There are four elements in the samples: Ti, C, O, and Cl. Figure 9a shows the wide spectra elements of the sample while Figure 9b represents the C1s spectra in which the C–C and C–O are located at 284.5 and 286.8 eV, respectively. Figure 9c illustrates the O1s characteristic peaks of Ti–O, C–O–C (aliphatic), and epoxide at 529.3, 532.5, and 533.1 eV respectively. The appearance of the peak of C–O–C (aliphatic) verified the presence of a crosslink reaction. This is due to the ether crosslink product consisting of C–O–C (aliphatic). Lastly, Figure 9d displays the Cl2p characteristic peaks that indicate the

existence of chloride from the PVC. Thus, it has been confirmed that the crosslink effect occurred after the photoetching process due to the presence of C–O–C (aliphatic).

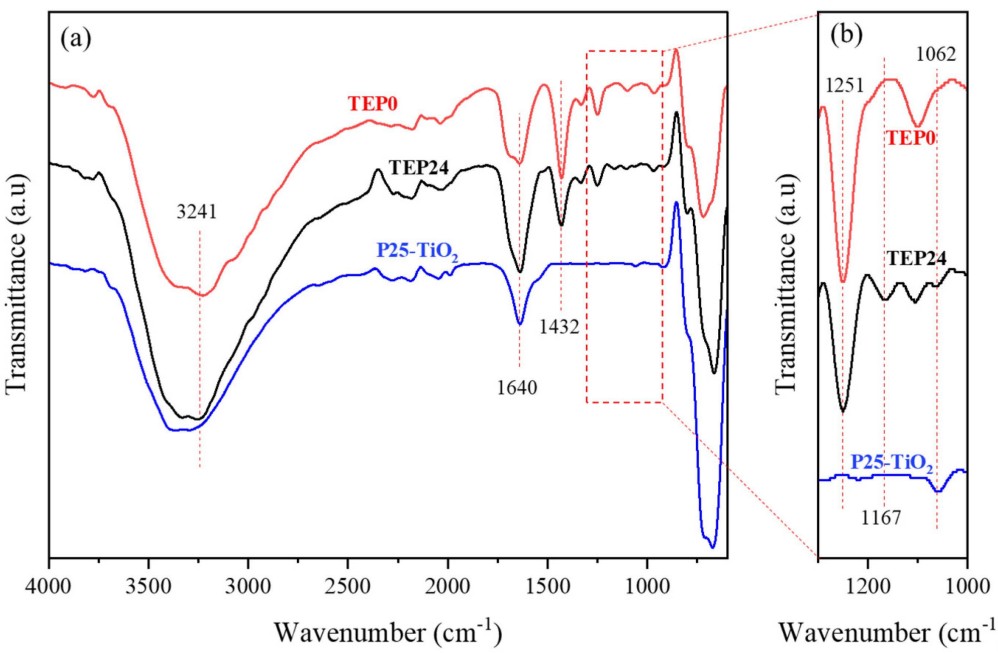

**Figure 8.** Comparison of Fourier transform infrared (FTIR) analysis between TEP0, TEP24, and degussa P25–TiO$_2$ in (**a**) a wide range wavenumber (4000–600 cm$^{-1}$) and (**b**) narrow range wavenumber (1300–1000 cm$^{-1}$).

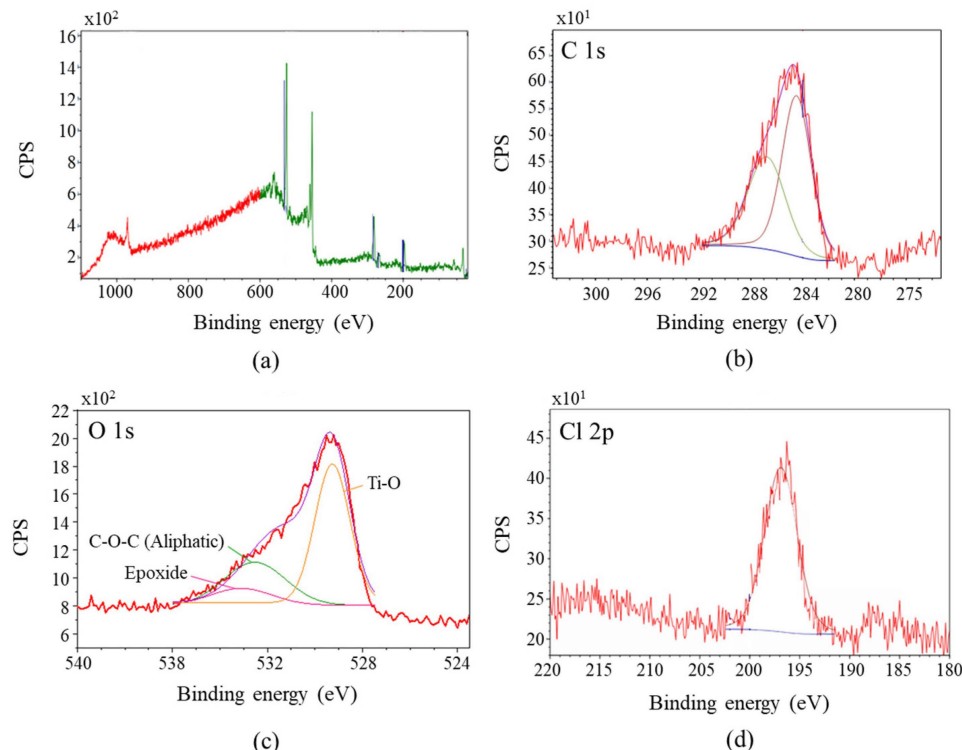

**Figure 9.** (**a**) The wide X-ray photoelectron spectroscopy (XPS) spectrum of (**b**) C1s (**c**) O1s, and (**d**) Cl2p for TEP24. CPS, cycles per second.

Based on the structure of ENR-50 in Figure 10, ENR-50 consists of an oxirane ring that is highly strained and readily open under mild conditions [48]. After the photoetching process, the oxirane ring in the ENR-50 structure was altered by a ring-opening reaction that formed radicals, as illustrated in the mechanism proposed in Figure 10. There are two possibilities of the ENR-50 crosslinking reaction as displayed in Figure 10(1,2). The opened ring of ENR-50 tends to react with PVC or another ENR-50, thus producing an ether crosslink product. The products derived from the crosslinking reaction may increase the durability and the adhesiveness of the immobilized $TiO_2$. Moreover, the radicals formed at the intermediate reaction (opening ring of ENR-50) can react with the RR4 dye (model pollutant) during the photocatalytic degradation as light penetrates towards the immobilized $TiO_2$/ENR/PVC, thus enhancing the photocatalytic degradation rate.

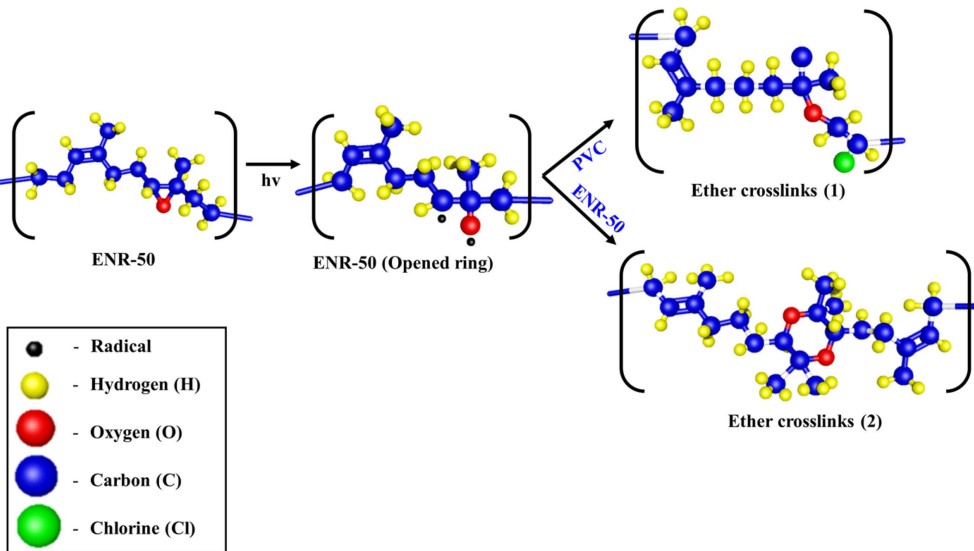

**Figure 10.** Possible reaction mechanism during the photoetching process.

The photoluminescence (PL) spectra of TEP0 and TEP24 were extracted to confirm the photoexcitation in the samples as shown in Figure 11. Basically, the PL intensity of TEP24 is quite higher than that of TEP0 at the sub-band range, which is around 500–700 nm wavelength. This situation has been expressed in the occurrence of electron hole recombination from the band to sub-band $TiO_2$ due to the superficial oxygen vacancies and flaws of the semiconductors. The photocatalytic activity of the TEP24 demonstrates a sophisticated response despite the high electron hole recombination since the photoetching process has created open pores through the internal sample, thus increasing the activation of the $TiO_2$ site.

*Photodegradation of Immobilized $TiO_2$/ENR/PVC*

The photodegradation of the immobilized $TiO_2$/ENR/PVC was carried out by using RR4 dye as a model pollutant. Figure 12 shows that the rate constant (k) value and percent dye removal increased with cycles of sample TEP24. The lowest rate constant (k) value was 0.0158 $min^{-1}$. After 24 cycles of photoetching, the rate constant (k) value reached the highest value of 0.1066 $min^{-1}$. The ENR from the immobilized $TiO_2$ was leached out by the photoetching process, thus increasing the surface area of the immobilized $TiO_2$ and improving the photocatalytic activity upon cycles.

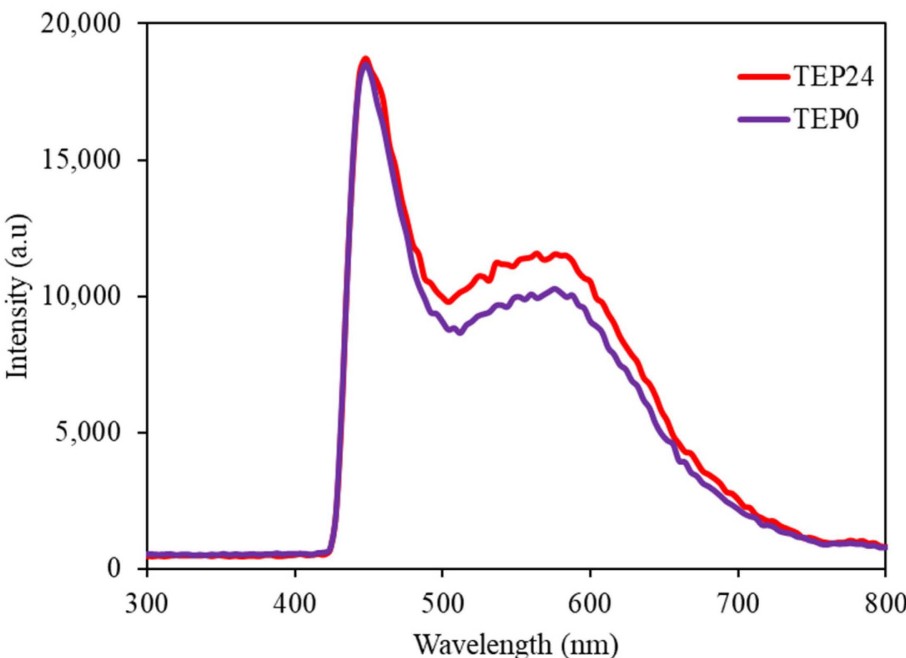

**Figure 11.** Photoluminescence (PL) spectroscopy of TEP0 and TEP24.

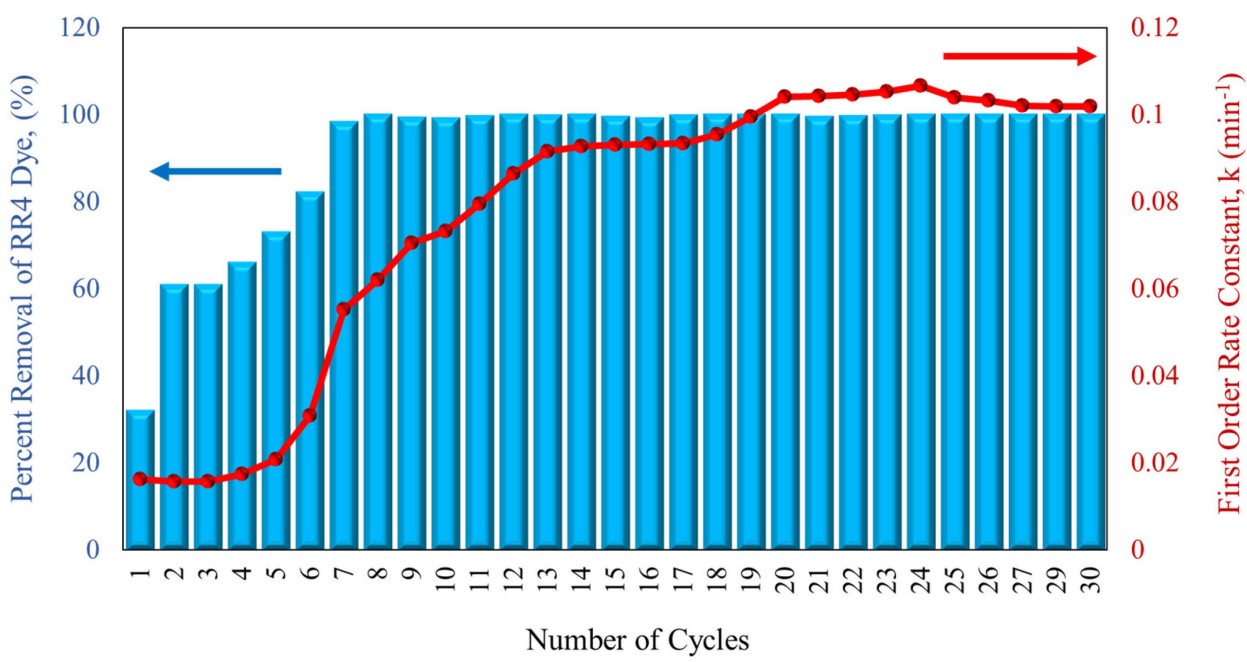

**Figure 12.** Pseudo first-order rate constant (min$^{-1}$) and percent removal of RR4 dye (%) vs. number of photocatalysis cycles using the TEP24 sample.

The photoetched immobilized TiO$_2$/ENR/PVC was about seven times faster as compared to the non-photoetched one, as shown in Figure 13. This is due to the opening ring and crosslinking effect and high contact area of the immobilized TiO$_2$. The ENR-50 compound on the immobilized TiO$_2$ surface was leached out during the photoetching process, thus increasing the contact area between TiO$_2$ and RR4 dye molecules [49]. However, the P25-TiO$_2$ suspension system was the uppermost, which is 8% better compared to the immobilized TiO$_2$, as it has the highest contact area.

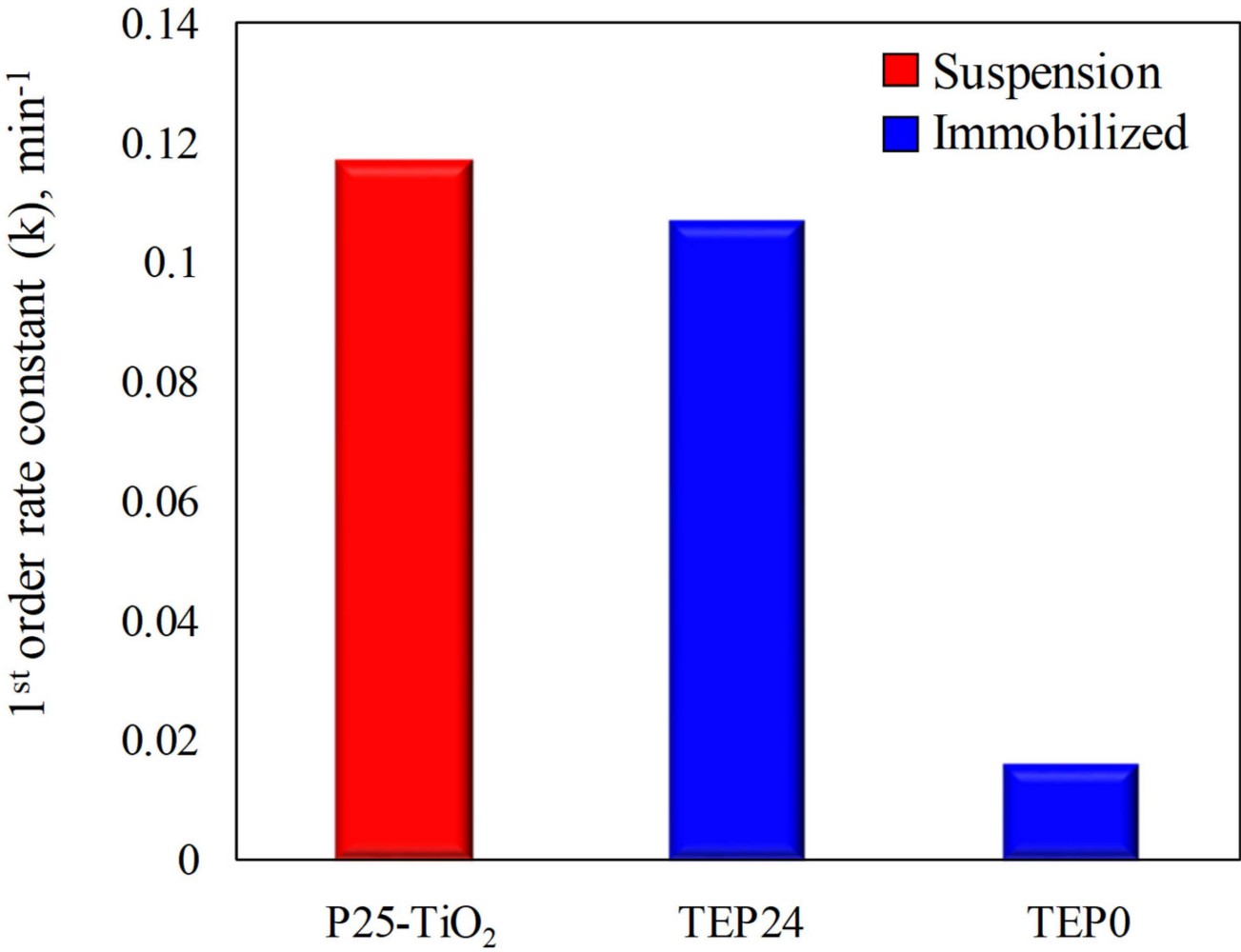

**Figure 13.** Pseudo first-order rate constant, (min$^{-1}$) for removal of 30 mg L$^{-1}$ RR4 dye via photocatalysis using P25–TiO$_2$ suspension system, and immobilized TEP0 and TEP24 catalyst plate.

Figure 14 shows the model of immobilized TiO$_2$ before and after the photoetching process. The porous immobilized TiO$_2$ was created due to the leaching of ENR from the immobilized TiO$_2$ during the photoetching process. Meanwhile, some of the ENR that remained in the immobilized TiO$_2$ underwent the ring-opening and crosslinking reactions as proposed in Figure 10. Despite the possibilities of increasing the durability and adhesiveness, the product of the crosslinking also may act as an electron injector to the TiO$_2$ as proven by the PL analysis in Figure 11, since the PL intensity for TEP24 is higher than for TEP0. Furthermore, some of the TiO$_2$ particles on the upper layer will be leached out together with the ENR during the photoetching process, thus activating the internal TiO$_2$ particles, which enlarge the contact area between the TiO$_2$ and dye solution. Therefore, the photoetching process can improve the photodegradation rate.

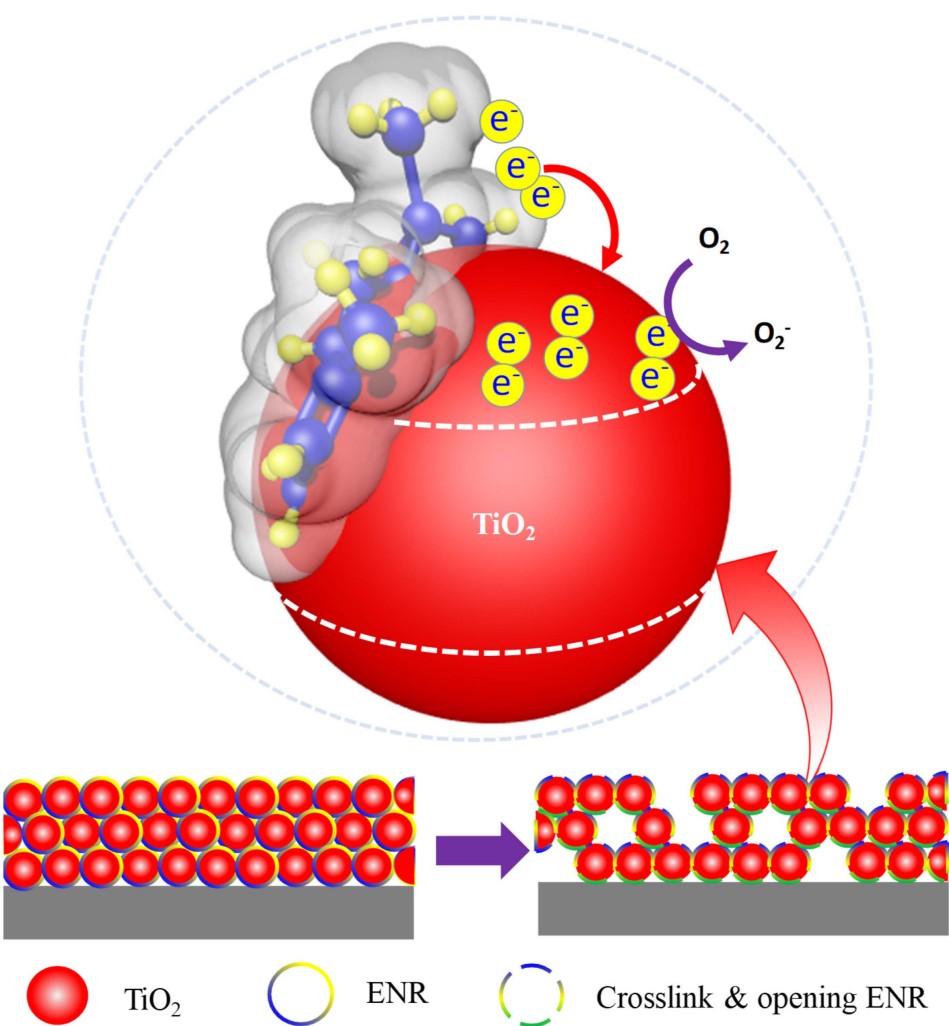

**Figure 14.** Diagram of immobilized $TiO_2$ surface before and after the photoetching process.

## 4. Discussion

This works represented the effect of a photoetching process on the surface of immobilized $TiO_2$/ENR/PVC and its reaction during the photoetching process. Photocatalytic degradation of RR4 dye by using a $TiO_2$/ENR/PVC plate under various times of the photoetching process has successfully enhanced the photocatalytic activity, making them able to degrade the dye solution into a colorless solution. This finding is significant since other immobilized catalyst may use the photoetching process to enhanced their photocatalytic activity. Moreover, this method can be applied to real textile wastewater in the future.

**Author Contributions:** Conceptualization, W.I.N. and S.R.H.; methodology, W.I.N., S.R.H. and M.A.R.; software, M.S.A., S.R.H. and N.S.N.; validation, M.A.M.I., R.N. and W.I.N.; formal analysis, M.S.A.; investigation, N.A.M. and S.R.H.; resources, M.A.M.I., W.I.N., M.S.A. and R.N.; data curation, W.I.N.; writing—original draft preparation, S.R.H.; writing—review and editing, W.I.N., N.S.N., M.A.R. and N.I.A.G.; visualization, W.I.N. and S.R.H.; supervision, W.I.N.; project administration, W.I.N.; funding acquisition, M.A.M.I. All authors have read and agreed to the published version of the manuscript.

**Funding:** This research was funded by the Ministry of Higher Education, Malaysia (MOHE), grant number 600-RMC/GIP 5/3 (006/2022).

**Data Availability Statement:** Not applicable.

**Acknowledgments:** The authors would like to thank the Ministry of Higher Education, Malaysia (MOHE) for providing financial support under GIP grants: 600-RMC/GIP 5/3 (006/2022), Universiti Teknologi MARA (UiTM) in conducting this study. We would also like to acknowledge Universiti Malaysia Perlis (UniMAP) and Universiti Teknologi MARA (UiTM) for providing all of the facilities.

**Conflicts of Interest:** The authors declare no conflict of interest.

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
