# Peer review of "The Crosslinking and Porosity Surface Effects of Photoetching Process on Immobilized Polymer-Based Titanium Dioxide for the Decolorization of Anionic Dye"

_2079-6447, doi:10.3390/colorants2010006_

Round 1

Reviewer 1 Report

The use of sample codes in the abstract is counterproductive as the reader would not understand what samples these are and miss the logic of the statements.

Line 45, replace adhesiveness by adhesion 

line 46, remove absorbance 

Line 72 and 73, explain why this particular choice

line 79, State the frequency of the irradiation

line 104, replace the phrase "prior to immobilizing the TiO2", with before the dip coating process. this makes it easier to read.

line 111-113; the difference in weight is actually the weight of the immobilized TiO2 and organic binder deposited to the glass plate.

line 114; state what was the typical number of cycles required to achieve the desired coating weight.

line 116; replace contained with filled.

lines 112 -130; did you use any internationally recognized testing method for this experiment? and if not explain why not. 

line 135. rephrase the sentence to make it readable; Figure 1 a and b shows that the TiO2 particles are covered with a layer of ENR/PVC.

Line 147 - 151; describe the method used i.e. stylus or optical probe?, probe dimensions, scanning rate, scanning area....

For the XRD analysis, This reviewer agrees with your interpretation however as the powder is likely made up is single crystals, you have to add the following for completeness sake. (assuming that the crystalline powder did not align itself to a preferential orientation during fixation), the strong diffraction peaks ....

181-182; This statement is wrong and needs to be corrected. the higher peak intensities are due to the etching of the polymer, exposing more TiO2 crystals to the X-Ray probe. The morphology of the ceramic powder  is unlikely to change under the reported experimental condition, other than some electrochemical etching.  

the difference in UV absorption intensity observed is also a consequence of the exposed TiO2 crystals after the binder was photo-etched with the very remote possibility that these might have been some preferential orientation of the crystals during curing and now different crystalline orientations are exposed to the UV rays.

figure 12. this shows the results for which sample?

Further relevant data missing.

1) what is the TiO2 particle loading in the coating?

2) How much g of organic material gets leached out in the water during etching?

3) What is the weight of TiO2 that is leached into the solution

Reviewer 2 Report

The manuscript presented by Hamzah and co-authors about The Crosslinking and Porosity Surface Effects of Immobilized TiO2/ENR/PVC via Photoetching Process for the Decolorization of RR4 Dye, the content looks interesting, however, the authors should show clearly the novelty for this study. As we know, the morphology is also important for the photocatalytic activity, and the authors have already added SEM images in the manuscript, but there are no such discussion, some discussion should be added. More details about Crystallinity index (CI) should be explained in the manuscript, the CI of TEP0 and TEP24 are 51.21% and 57.30%, respectively, are these results over accuracy? The format and quality of some figures could be better (e.g., Figure 2 and Figure 9, there are several styles for the Figures in this manuscript, font size and line thickness are quite different for different figures, the styles for these figures should be modified to be unified, and thus could be better for readers). Some sentences were written not so well, the English should be improved, and there are some small mistakes could be found in the content and also in the references, the mistakes should be corrected. All the references should be unified and followed the requirements of this journal, the authors should check these carefully.  

Reviewer 3 Report

The authors used photoetching to investigate the crosslinking and porosity surface effects of immobilized TiO2/ENR/PVC for the decolorization of RR4 dye. I feel that this work seems interesting. This paper may be recommended for publication; however, the following revisions need to be made before publication.

  1. The novelty and originality of this study should be strengthened.
  2. The purity of all chemicals used in the study should be provided.
  3. It is better to add the detailed characteristics of Degussa P-25 TiO2 in XRD, FTIR, UV-Vis DRS, etc.
  4. The mechanism part needs to be improved by adding references.

Round 2

Reviewer 1 Report

The current manuscript is much better 

Author Response

Point 1: The current manuscript is much better.

Response 1: Thank you very much. It is a great pleasure to have this comment.